# Anticholinergic Burden, Sleep Quality and Health Outcomes in Malaysian Aged Care Home Residents

**DOI:** 10.3390/pharmacy7040143

**Published:** 2019-10-23

**Authors:** Suresh Kumar, Syed Shahzad Hasan, Pei Se Wong, David Weng Kwai Chong, Therese Kairuz

**Affiliations:** 1Department of Pharmacy Practice, International Medical University, Kuala Lumpur 57000, Malaysia; peise_wong@imu.edu.my (P.S.W.); DavidChong@imu.edu.my (D.W.K.C.); 2Department of Pharmacy, University of Huddersfield, Huddersfield HD1 3DH, UK; s.hasan@hud.ac.uk; 3School of Biomedical Sciences and Pharmacy, University of Newcastle, Callaghan, NSW 2308, Australia

**Keywords:** anticholinergics, ADL, frailty, quality of life, sleep, aged homes

## Abstract

The use of anticholinergic medications by residents in aged care homes is associated with increased risk of adverse effects. These include cognitive impairment, sleep disturbances, and falls, and necessitate increased healthcare visits and the associated burden on healthcare systems. The objective of this study was to investigate associations between anticholinergic burden and health outcomes such as independence in activities for daily living, frailty, quality of life, and sleep quality. The study was conducted among residents in Malaysian aged care homes, aged 60 years and above. Anticholinergic burden was calculated using the Anticholinergic Cognitive Burden (ACB) scale. Health outcome measures included independence, assessed using the Katz Activities for Daily Living scale (Katz ADL); quality of life, assessed using the Older People’s Quality of Life Questionnaire (OPQOL); frailty, assessed using the Groningen Frailty Index (GFI); and sleep quality, measured using the Pittsburg Sleep Quality Index (PSQI). Just over one-third (36%) of the study population was exposed to at least one medication with anticholinergic effect. An increased anticholinergic cognitive burden was associated with frailty (*p* = 0.031), sleep latency (*p* = 0.007), and sleep disturbances (*p* = 0.015). Further studies are required to assess the effect of prolonged exposure to anticholinergic medications on health outcomes.

## 1. Introduction

Malaysia is a multi-ethnic country with a population of 32 million (in 2017) and an expected annual growth rate of 1.1% [1]. Older people comprise nearly one-tenth of the Malaysian population (8.8%) [2], and the proportion of people aged ≥ 65 years is projected to reach 7.2% in 2020 and to grow by more than double by 2040 [1]. Increasing numbers of frail elderly people with multiple morbidities will result in a high demand for long-term care facilities [3].

Older people may require multiple medications, predisposing them to polypharmacy. It has been reported that at least one medication is likely to have anticholinergic and/or sedative properties [4]. Drugs with anticholinergic effects may cause dizziness, drowsiness, sedation, confusion, memory impairment, blurred vision, cognitive impairment, and increased risk of falls due to the action on acetylcholine. Drugs with anticholinergic properties, such as diuretics and stimulants, are known to cause sleep disturbances, whereas sedatives may lead to daytime drowsiness. These effects of polypharmacy may negatively impact sleep quality and quality of life [5]. The term “sleep quality” includes sleep latency (time taken to fall asleep after going to bed), sleep efficiency (percentage of time spent asleep in bed) [6], sleep duration (total amount of sleep obtained) [7], as well as sleep disturbances (disorders of initiating and maintaining sleep) [8] and the resultant daytime dysfunction (lack of enthusiasm and trouble staying awake) [9]. Exposure to anticholinergic medications is associated with cognitive impairment, dizziness and lack of coordination, sleep disturbances and daytime dysfunction, and urinary retention [4,10,11,12]. A combination of various effects results in a high incidence of falls and an increased risk of fractures in older adults [10,13].

Frail older adults are often characterized by multiple morbidities and mortalities, and risk is exacerbated. Frailty is a geriatric syndrome characterized by a decline in physiological functions as well as homeostatic reserve [14]. It is an indicator of adverse health outcomes, and frailty is considered to be a better predictor of fluctuations in health risk than age [15,16]. Older adults exposed to high anticholinergic burden are at greater risk of adverse health events due to the risk of falls and fractures. In addition to the previously listed adverse effects, health outcomes associated with anticholinergic effects include cognitive impairment, lack of coordination, depression and sleep disturbances, and urinary retention [4,11,12]. These effects are compounded by cumulative anticholinergic cognitive burden [17]. High anticholinergic cognitive burden and the associated loss of cognitive function impact reduced physical function and activities for daily living [18].

Katz et al. developed an index of activities for daily living (ADL) that can be used to classify the functional status of older adults. Functional status is an important measure of health (and disease) and indicates the requirements for, and outcomes of, health service provision. Katz’s ADL index measures a person’s ability to perform daily tasks independently and includes bathing, dressing, toileting, incontinence, feeding, and transferring [19]. The ADL index serves as a predictive measure of chronic diseases, and may be used to assist in the process of measuring health needs and health outcomes. In turn, needs and outcomes can be used for the purposes of planning, policy-making, research, and teaching [20].

Despite the evidence of adverse effects from these medicines, the use of prescription and non-prescription medications with anticholinergic effects is increasing among older people [17], including treatment for pain, psychological conditions, urinary incontinence, and airway disease [21]. Increased exposure to anticholinergic drugs in older adults is also associated with reduced quality of life [22,23,24,25]. Therefore, it is important to quantify the effects of medications with anticholinergic effects, particularly among vulnerable older adults. The primary aim of the current study was to determine the correlation between anticholinergic cognitive burden and sleep quality, activities of daily living, frailty, and quality of life, to rationalize the safe use of medicines and inform health policy. The first objective was to describe the number of anticholinergic medications in a sample of older adults in aged care homes in Malaysia, and the anticholinergic cognitive burden associated with their medications. The second objective was to identify a relation between anticholinergic cognitive burden (score) and health outcomes, namely, independence in activities of daily living, quality of life, frailty, and sleep quality.

## 2. Materials and Methods

### 2.1. Study Design, Setting, and Participants

Data for this cross-sectional study were collected from 11 aged care homes in the states of Selangor, Kuala Lumpur and Ipoh in Malaysia, between July and September 2017. A sample size of 133 (confidence interval of 95% and a margin of error of 4%) was calculated based on the size of the elderly population in Malaysia (5.89%) at the time [1,26,27]. Statistical evaluation was conducted using Chi-Square and one-way ANOVA to measure the differences in means among exposed and non-exposed groups. Correlation between Anticholinergic Cognitive Burden score and health outcomes (independence in activities of daily living, frailty, quality of life, and sleep quality) was tested using Spearman’s rank correlation coefficient. All analyses were completed using SPSS version 25.

All residents aged > 60 years, who were taking/using at least one prescribed or over-the-counter (OTC) medication were eligible to participate. OTC includes non-prescription medicines; Chinese medicines, homoeopathic remedies, vitamins, and supplements were excluded, as were residents who had a diagnosis or known history of dementia and sleep disorders. Data were collected using an interview-administered assessment form; this questionnaire included demographic data of the study participants, questions assessing activities of daily living, quality-of-life of older adults, frailty, and sleep quality. Data on medical conditions and medications were obtained from the medication records and from personal history files maintained in the aged care homes. Data were anonymized to maintain the confidentiality of individuals, and written informed consent was obtained from participants to participate in this study and to access their medication profiles.

Ethical approval was granted by the International Medical University Joint Committee for research and ethics (Project ID no. IMU 385/2017).

### 2.2. Outcome Measures

The primary outcome of this study was the proportion of older adults exposed to drugs with an anticholinergic cognitive burden, and its relationship with sleep quality. Secondary outcomes included the association between anticholinergic burden and health outcomes such as ADL, quality of life, and frailty.

Anticholinergic cognitive burden was calculated using the Anticholinergic Cognitive Burden (ACB) scale, which classifies medications according to negative cognitive effects. Medications that could exhibit serum anticholinergic activity without clinically significant anticholinergic effects were assigned an ACB score of 1 (“possible anticholinergics”), while those with clinically relevant anticholinergic effects scored 2 or 3 (“definite anticholinergics”) [28,29]. The total anticholinergic score for each participant was calculated by adding the individual ACB scores of drugs taken by the corresponding individual.

Activities for daily living were scored according to the Katz ADL, which is the most widely used tool for measuring the functional status of older adults [19,20]. It comprises six activities categorized as either “independent” or “dependent”; the total score ranges from 0 (full independence) to 6 (full dependence).

Quality of life was measured using the Older People’s Quality of Life (OPQOL) questionnaire [30]. OPQOL comprises 35 items in 8 domains: life overall, health, social relationship, independence, home, psychological, financial, and leisure. Each item has a five-point Likert scale, with higher scores indicating better QoL. Total OPQOL score ranges from 35 (worst possible) to 175 (best possible) [31].

Frailty was evaluated using the Groningen Frailty Index (GFI), which is a validated tool to measure frailty among institutionalized older adults [32]. The GFI comprises fifteen dichotomous items with a score ranging from 0 (normal activity without restriction) to 15 (completely disabled, very weak, unable to carry out any self-care activities, and bedridden). Subjects with a score of ≥4 are considered frail [33].

Sleep quality was measured by using the Pittsburgh Sleep Quality Index (PSQI). The PSQI instrument is frequently employed to assess subjective sleep quality, and differentiates between good and poor sleepers. PSQI uses 19 self-rated questions which are scored according to seven components, each with scores 0–3 and weighted equally. Seven components of the PSQI instrument measure global sleep quality score, namely, sleep quality, sleep latency, sleep duration, habitual sleep efficiency, sleep disturbances, use of sleep medications, and daytime dysfunction. The Global PSQI score is the sum of individual components and ranges from 0 to 21, where higher scores indicate poorer sleep quality; subjects with global PSQI scores ≥ 5 are considered to have poor sleep quality. Psychometric properties of the PSQI instrument are reliable, with internal consistency reliability ranging from 0.80 to 0.88 and test–retest reliability from 0.85 to 0.87.

The PSQI has good sensitivity and specificity in subjects with or without sleep problems when estimated at a cut-off score of 5 [34,35,36,37]. To differentiate between participants with “poorer” sleep quality (i.e., those with moderately poor sleep quality and very poor sleep quality), we used a second cut-off at 10 [38].

## 3. Results

There were 151 residents who met the inclusion criteria and consented to participate. The demographic and health-related characteristics of the study population are shown in Table 1.

### 3.1. Exposure to Anticholinergic Medications

Just over one third (36%, n = 151) of older adults received at least one anticholinergic medication (average of 0.44, SD = 0.64). The mean ACB score of the study population was 0.60 (SD = 0.99). Among the participants exposed to anticholinergics, two-thirds (64%) were using a medication with “possible” anticholinergic effect (ACB score = 1) and approximately one third (36%) were using (at least one) “definite” anticholinergic medication (ACB score = 2–3). A list of drugs used by the study population, classified according to the ACB score, is provided in Appendix A.

### 3.2. Health Outcomes

Most participants (79%) were able to carry out activities of daily living independently, and the mean Katz ADL score was 0.99 (Table 2). 

### 3.3. Health Outcomes by Anticholinergic Cognitive Burden 

Health outcomes according to anticholinergic exposure was assessed using the Mann–Whitney test. Except for frailty (*p* = 0.031), health outcomes did not show a significant difference in mean between exposed and non-exposed groups (*p* > 0.05) (Table 3).

Frailty was the only health outcome that was significantly correlated with the Anticholinergic Cognitive Burden (*p* = 0.031) (Table 4).

### 3.4. Sleep Quality

Associations between sleep quality and anticholinergic cognitive burden were tested using five components of overall sleep quality: sleep latency, sleep efficiency, sleep disturbances, daytime dysfunction, and sleep duration. Sleep latency was found to be negatively correlated with the Anticholinergic Cognitive Burden, and there was a significant association between sleep latency (*p* = 0.007, correlation coefficient = −0.220) and sleep disturbances (*p* = 0.015, correlation coefficient = 0.198) (Table 5).

## 4. Discussion

The current study evaluated the use of anticholinergic medicines among older adults residing in aged care homes in Malaysia. The study also quantified associations between Anticholinergic Cognitive Burden and physical health outcomes such as independence, frailty, and sleep quality. About one-third of the study population were exposed to at least one anticholinergic drug, which supports the findings of similar clinical studies [39,40,41]. They were able to carry out their activities of daily living, and more than half of them reported a good quality of life. However, most of the older adults were frail and reported poor sleep quality.

High anticholinergic use could be associated with specific study populations with a high prevalence of conditions such as depression and Parkinson’s disease, as they often require treatment with one or more medications which have anticholinergic effects. A notable exception is the recent (2019) KORA-FF4 study in Germany, which reported a low exposure to anticholinergics. However, KORA-FF4 was conducted among community-dwelling elderly who had fewer comorbidities than subjects in the current study [42]. Just over one-tenth (13%) of our subjects were prescribed anticholinergics that had clinically significant anticholinergic effects/side effects (Katz ADL score 2–3); the medications were mainly to treat Parkinsonism, psychiatric conditions, sleep disturbances, and gastrointestinal disorders.

### 4.1. Effects on Physical Function

Findings from our study suggest that an increase in frailty is associated with an increase in anticholinergic burden. It supports literature arising from the Concord Health and Ageing in Men Project Cohort Study, and the InCHIANTI study. Both reported an increase in anticholinergic burden during the transition to frail state [43,44]. Increased exposure to anticholinergic medications may contribute to the deterioration of physical functions that characterize frailty [45]. This association may be due to the presence of comorbidities in frail individuals, which requires the use of many medications with anticholinergic properties. Side effects may be due to pharmacological actions, including drowsiness, dizziness, cognitive and functional impairments [46], and may be more pronounced due to altered physiology associated with ageing processes. In our study, independence in activities of daily living and health outcomes such as quality of life were not associated with Anticholinergic Cognitive Burden score; however, other studies have demonstrated a positive association between Anticholinergic burden and these parameters [47,48]. Studies conducted in the USA, the Netherlands, New Zealand, and Italy found that increased exposure to medications with anticholinergic effects is associated with a decline in physical function, cognitive impairment, and quality of life among older adults [22,23,24,25].

### 4.2. Effects on Sleep Quality

Anticholinergic effects on the central nervous system (drowsiness, dizziness, sleep disturbances, and cognitive impairment) [49,50] may affect sleep quality in older adults. To the best of our knowledge, the current study is the first to report an association between subjective sleep quality and Anticholinergic Cognitive Burden. Furthermore, associations between individual components of sleep quality and Anticholinergic Cognitive Burden were significantly associated with sleep latency (negative correlation). It was evident that subjects with a higher anticholinergic cognitive burden spent less time in bed before they fell asleep when compared to those who were not exposed to medications with anticholinergic effects. This “negative” association may be due to central nervous system side effects in this population, and could be beneficial as sleep latency increases with age [51]. However, depression of the central nervous system may lead to daytime drowsiness and an increased risk of falls and fractures.

Anticholinergic cognitive burden was positively associated with sleep disturbances, consistent with findings from other studies [49,50]. However, we did not find an association between overall sleep quality, sleep efficiency, sleep duration, and daytime dysfunction. According to pharmacologic profile, anticholinergics would be expected to affect sleep duration. The absence of an association may be due to the relatively short duration of exposure to anticholinergic medication(s), or to individuals’ perceptions about their sleep efficiency. Further studies are recommended to assess sleep quality and the duration of anticholinergic exposure, using subjective measures (e.g., PSQI) as well as objective measures (e.g., actigraphy), using a longitudinal study design.

### 4.3. Limitations

It is possible that relationships were not established because potential sedative effects were excluded and only anticholinergic cognitive burden was considered. Another factor that may have contributed to non-association is the duration of exposure, which was limited given the cross-sectional study design. In addition to the duration of exposure and characteristics of the study population (see above), we used a subjective sleep quality measure; actual sleep quality may not be reflected, as it is influenced by individual perception.

## 5. Conclusions

In this cross-sectional study among a sample of aged care residents in Malaysia, the anticholinergic cognitive burden was correlated with frailty, sleep latency, and sleep disturbances. The findings highlight the importance of reducing anticholinergic cognitive burden among older adults, thereby minimizing adverse health outcomes.

## Figures and Tables

**Table 1 pharmacy-07-00143-t001:** Characteristics of participants. ACB: Anticholinergic Cognitive Burden.

Variable	n = 151	%
Age in years, Mean(SD)	74.47 (8.30)	
**Gender**,		
Male	77	51
**Marital Status**		
Married	50	37.6
Single/divorced/separated	83	62.4
**Education**		
≤Primary (0–6 years)	77	53.5
Secondary (7–11 years)	59	41
Tertiary (diploma or degree and above)	6	4.2
**ACB category**		
Exposure to ACB drugs	55	36.4
Exposed to Possible anticholinergic		
(ACB score = 1)	35	23.2
Exposed to definite anticholinergic		
(ACB score ≥ 2)	20	13.3
**Chronic Conditions**		
Cardiovascular diseases	101	66.9
Metabolic diseases	94	62.3
Psychiatric conditions	18	11.9
Respiratory diseases	9	6
Benign prostatic hyperplasia	11	7.3
Parkinson’s disease	8	5.3
Gout	3	2

**Table 2 pharmacy-07-00143-t002:** Health outcome scores of study participants. ADL: Activities of Daily Living; OPQOL: Older People’s Quality of Life; PSQI: Pittsburgh Sleep Quality Index.

	ACB Score	Katz ADL	OPQOL Total	GFI Score	Global PSQI
Mean	0.60	0.99	109.84	4.56	9.68
Median	0	0	110.00	5.00	9.00
Std. deviation	0.994	2.013	8.053	2.710	2.647
IQR	0–1.00	0	105.00–114.75	2.00–6.00	8.00–12.00
Minimum	0	0	91	0	4
Maximum	4	6	142	13	15
Possible range	-	0–6	35–175	0–15	0–21

Regarding sleep quality, the majority (93%) of the study population had poor sleep quality (PSQI > 5); the mean PSQI score was 9.68. Frailty (≥ 4) was estimated using the Groningen Frailty Index (GFI), and the average GFI was calculated as 4.56; 75% of the population were frail. Quality of life was low; just over half (52%) reported poor quality of life, with a mean OPQOL score of 109.8 (range 35–110).

**Table 3 pharmacy-07-00143-t003:** Health outcomes by Anticholinergic Cognitive Burden (ACB) score.

	ACB Category	n *	Mean (SD)	*p*-Value
Katz ADL	No Exposure	96	0.98 (2.016)	0.951
	Exposure to ACB drugs	55	1.00 (2.028)	
OPQOL Total	No Exposure	93*	109.47 (7.902)	0.467
	Exposure to ACB drugs	55	110.47 (8.337)	
GFI Score	No Exposure	96	4.20 (2.721)	0.031
	Exposure to ACB drugs	55	5.18 (2.597)	
Global PSQI	No Exposure	96	9.70 (2.664)	0.891
	Exposure to ACB drugs	55	9.64 (2.641)	

* Three participants did not complete the quality-of-life component.

**Table 4 pharmacy-07-00143-t004:** Correlation between Anticholinergic cognitive burden and health outcomes.

Variables		ACB Score	
	n	Correlation Coefficient	*p*-Value
Katz ADL	151	0.000	1.000
OPQOL Total	148 *	0.104	0.209
GFI Score	151	0.169	0.039
Global PSQI	151	0.006	0.942

* Three participants did not complete the quality of life component.

**Table 5 pharmacy-07-00143-t005:** Correlation between Anticholinergic Cognitive Burden and sleep quality components.

Sleep Quality Component		Total ACB Score	
	n	Correlation Coefficient	*p*-Value
Sleep latency	151	−0.220	0.007
Sleep efficiency	151	−0.008	0.921
Sleep disturbances	151	0.198	0.015
Sleep duration	151	0.013	0.870
Day time dysfunction	151	0.058	0.482

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
