# Peer review of "Anticholinergic Burden, Sleep Quality and Health Outcomes in Malaysian Aged Care Home Residents"

_pharmacy, 2019, doi:10.3390/pharmacy7040143_

Round 1

Reviewer 1 Report

The authors investigated the correlation between the anticholinergic cognitive burden (ACB) and quality of sleep score, frailty score, and quality of life. They found that the anticholinergic cognitive burden was negatively correlated with sleep latency and positively correlated with sleep disturbance and frailty. They did not observe significant correlations between ACB and other outcomes.Generally, the authors may have to use statistics appropriate for ordinal scores for accurate summaries. 

Comments:

Given that variables are ordinal score types, Spearman's rank based correlation instead of Person's correlation is recommended. In table 1, the mean and standard deviation can be provided in the same column as mean(SD) thus eliminating the need for an additional column.  In table 1, the category for education does not include primary as the levels include <primary, secondary and tertiary. The definition of ACB included a score of 1 if medications exhibited serum anticholinergic activity without clinically significant anticholinergic effects. Did you study the blood samples of the patients to determine whether a given medication has serum anticholinergic effect? Is the anticholinergic cognitive burden (ACB) score given per drug or person? This is not clear. The summary results discuss the summary statistics of ACB scores in the study population. Please make the definition of ACB score clear.   Please use the equal sign between SD and the number not a hyphen (SD=0.99) for ACB. You may also use, the mean(SD) score of ACB was 0.60(0.99). I generally recommend using the median and interquartile range for ordinal scores. In table 2, given that the outcome measures are scores, it is better to use the median and interquartile range and not the mean and standard deviation.  In table 3, the horizontal lines under the row names are not necessary. Whether the negative correlation between the ACB and sleep latency is a positive effect or negative/adverse effect of the ACB is not well described. Spending less time in bed before falling asleep is thought of as positive. The authors need to comment on that.   The authors did not describe how they obtained the p-value in table 3. I recommend changing the summary into median (IQR) and compare binary categories using Wilcoxon-rank test for example.

Reviewer 2 Report

The concept of finding a relationship between anticholinergic burden and sleep is interesting and this study builds on work done in other settings over the past few years.

Areas to address in this article:

Abstract - may benefit from including a descriptor or two about the population (e.g. the n, not just %; female, age)

Keywords - please confirm if these are MeSH headings

Throughout - change 'elderly' to 'older adults'

Introduction - second paragraph moves suddenly from polypharmacy to sleep - please link these concepts.  

Methods 

Please clarify what is the primary outcome

Was the n calculated based on a particular outcome?  How did you arrive at this sample size?

Were the data records from the pharmacy records or the national profile?

Please clarify if 'disabled' meant frail, or how you defined disabled?

Did subjects have any autonomic dysfunction?  Any other measures to control for?

Results

Please define education by primary, secondary, tertiary, as these terms vary internationally

Please expand your description of the population in table 1 - other comorbidities, such as depression, Parkinson disease, over-the-counter use, history of sleep aids, etc.  I don't see current use of any sedating medications listed.

Please list the number of high risk medications as well as the number of medications overall in table 1

Table 2 - assume that all findings were normally distributed, and no need for median or other measures?

Table 5 - suggest confirming that each component being tested is valid to be tested independent of the larger tool.  Sometimes over-testing many times leads to statistical significance.

Discussion - please re-state your findings or a summary of them in the opening paragraph.

Effects on physical function - this opening statement isn't what you found - it seems unrelated to the study, as you were focusing on sleep.

Bottom line - I'm not sure the study design allows the conclusions.  If you did not control for benzo or other sedating drug use, or control for conditions that affect sleep or function, the results are only hypothesis generating.

Round 2

Reviewer 1 Report

The authors have made the requested changes and made a thoughtful revision.

In table 2, the authors can present an IQR of zero as 0; no need to put 0.00-0.00. 

Author Response

Dear Reviewer

As recommended, we have done a thorough spelling check and done some minor edits such as spacing. Please see the below response to your comment.

Comment

In table 2, the authors can present an IQR of zero as 0; no need to put 0.00-0.00. 

Response: In table 2, Median and IQR for both ACB score and Katz score have been amended from "0.00" to "0".

All changes are done using track changes. 

Thank you

Regards

Suresh Kumar